# Thermal Shock Response of Yeast Cells Characterised by Dielectrophoresis Force Measurement

**DOI:** 10.3390/s19235304

**Published:** 2019-12-02

**Authors:** García-Diego Fernando-Juan, Mario Rubio-Chavarría, Pedro Beltrán, Francisco J. Espinós

**Affiliations:** 1ACUMA Research Centre Universitat Politècnica de València, Av. de los Naranjos s/n, 46022 Valencia, Spain; pbeltran@fis.upv.es (P.B.); fespinos@acuma.upv.es (F.J.E.); 2Escuela Técnica Superior de Ingenieros Industriales Universitat Politècnica de València, Av.de los Naranjos s/n, 46022 Valencia, Spain; marucha1@etsii.upv.es

**Keywords:** DEP, dieletrophoretic force, Stokes, thermal, yeast cells, dieletrophoresis, differentiation

## Abstract

Dielectrophoresis is an electric force experienced by particles subjected to non-uniform electric fields. Recently, several technologies have been developed focused on the use of dielectrophoretic force (DEP) to manipulate and detect cells. On the other hand, there is no such great development in the field of DEP-based cell discrimination methods. Despite the demand for methods to differentiate biological cell states, most DEP developed methods have been focused on differentiation through geometric parameters. The novelty of the present work relies upon the point that a DEP force cell measurement is used as a discrimination method, capable of detecting heat killed yeast cells from the alive ones. Thermal treatment is used as an example of different biological state of cells. It comes from the fact that biological properties have their reflection in the electric properties of the particle, in this case a yeast cell. To demonstrate such capability of the method, 279 heat-killed cells were measured and compared with alive cells data from the literature. For each cell, six speeds were taken at different points in its trajectory inside a variable non-uniform electric field. The electric parameters in cell wall conductivity, cell membrane conductivity, cell membrane permittivity of the yeast cell from bibliography explains the DEP experimental force measured. Finally, alive and heat-treated cells were distinguished based on that measure. Our results can be explained through the well-known damage of cell structure characteristics of heat-killed cells.

## 1. Introduction

Nowadays, the number of technologies centred on the use of dielectrophoretic force (DEP) is awe-inspiring. These methods have numerous applications in fields such as biochemical analysis [1] and particles manipulation applications that could be grouped according to the object of control: cells, nanoparticles and viruses. 

In cells manipulation, DEP has been employed for many purposes, such as cell guidance inside lab-on-a-chip devices [2,3], cell isolation [4] or as a basis for label-free techniques of sorting and separation of stem cells and their differentiation products [5]. In nanoparticle manipulation, DEP has been utilized in many DNA-oriented technologies. With applications such as handling [6], trapping [7], recovering from plasma [8] or cell free circulating DNA (cfc-DNA) isolation [9]. Related to virus manipulation, DEP is used for capture and separation from soil particles [10] among other applications.

In conjunction with the development of bioinformatics and gene sequencing technologies, many of these manipulation techniques have been applied in the study of cancer cells [11,12,13], specifically circulating tumour cells (CTCs). Furthermore, closely related to the study of CTCs, there is another branch of DEP-based technologies devoted to the detection of cancer cells [11,14], viruses [15,16] and nanoparticles [9,17,18]. The potential of all these DEP-based technologies rely upon their sensitivity, specificity [11,19,20,21,22,23], cheapness and ease of use [24]. 

Conversely, over these last two decades, another potential application of DEP has been set aside: differentiation. In this paper, the term differentiation refers to the activity of determining the nature or type of particles, generally cells, based on their properties. Due to the fact that the principle employed is DEP, most of the time electric properties are employed as differentiation features. 

One advantage of DEP against other differentiation techniques is that it is the basis of a group of label-free techniques [2,3,12,24,25,26,27,28]. Nevertheless, an increasing number of works are published every year in this field. Specifically, DEP-based differentiation has mainly been used to study nanoparticles [21,24] and cell discrimination [23,25,26,27,28,29,30]. 

In previous works [31,32,33], a method of cell discrimination was developed through direct measurement of DEP force. This theoretical model is based on Newton’s Second Law, assuming that the forces involved in a cell in a non-uniform electric field are DEP, viscous friction and the cell’s weight in the fluid. At this point, it is supposed that the mass by acceleration term is negligible compared with the other forces involved. Thus, by the measure of the cell velocity in the fluid, a parameter, which quantifies the force exerted by a potential difference of one volt on the particle (*F_1_*) was measured. This parameter depends on the shape of the particle, the medium viscosity and dielectric features of the particle and medium. 

However, all of these papers [31,32,33] ground their reasoning on a geometric basis. In other words, different cell forms involve different polarisation and different drag resistor forces. In [31], the *F_1_* distribution was obtained in a population of (*Saccharomyces cerevisiae*) yeasts cells and compared with the radius distribution. In [32], the model is validated for concave particles (red blood cells). In the third, [33], the same is achieved for single and double yeast cells. All three articles applied the method that is exposed here, but conversely with which is treated in this article, they all were geometrically oriented.

Nonetheless, different cells may present similar shapes but differ in other biological features, such as inner cell structure or cell wall composition. Such features are represented in dielectrophoretic cell properties and, therefore, are reflected in the cell’s DEP response. In this work, heat-killed yeast cells are studied. This heat shock supposes changes in conductivities of the cell wall and membrane, and cell membrane permittivity with respect to the cells of the alive group [33]. These changes provoke a distinct reaction to DEP force, conferring different measured velocities in cells.

Therefore, these features constitute potential DEP-based criteria for cell differentiation. According to it, among the biologically-oriented cell discrimination methods there is none based on DEP measurement through direct application of Newton’s Second Law and Stokes drag force. This article aims to fill this gap. We demonstrate that, despite being based on geometric cell characteristics [31,32,33], this method is valid for the discrimination of different cell biological states. 

Moreover, regarding DEP applied to yeast cells, state of the art hinges around microfluidics and cell manipulation [34,35,36,37,38,39]. In cell preparations, there is concern about overheating the samples [34,40], and, as claimed before, most of the newest discrimination methods are built upon geometric principles [35]. Therefore, there is a lack of cell discrimination methods based on biological features of the cell. In this article the method that has been developed in [31,32,33] will be used to perform this discrimination. In particular, it has been applied to discern between cells damaged by heat treatment and the alive ones [33]. Admittedly, other approaches have been employed to the same problem [41], though none of them to yeast cells. Consequently, finding yeast cell discrimination methods applicable to determine biological cell parameters, in this case, the damage caused by heating, reveals itself as an objective worth pursuing.

In the previous experiment [33], it was verified that this method was sensitive in detecting variations in cell wall conductivity, cell membrane conductivity, cell membrane permittivity and cell shape. It was achieved for both single and double cells. These changes in conductivity and permittivity may be caused by thermal treatments, ultrasounds or the effect of detergents [42,43,44,45]. 

The aim of this work is to demonstrate that these changes, well documented in [43,46], could be measured through the same technique described in [33]. The ease of performing the treatment and the abundance of [43,46] were the two main reasons why we decided to perform a thermal treatment among the three possible alterations mentioned above. In this experiment, the same cell preparation and the same chamber were used as in the previous assay [33].

## 2. Theoretical Model

As described above, this work follows the same methodology as previous works [31,32,33]. 

In brief, the cellular trajectories are located in the XY plane and only cells with vertical trajectories along the *y*-axis were chosen for the measurements (Figure 1). Applying Newton’s Second Law:$$\sum^{\text{​}}F=m\cdot a$$
$$F_{sto}-F_{g}-F_{DEP}=m\cdot a$$

The forces exerted over each cell are the followings:

$F_{g}$: The effective weight of the particle in the medium (where $\rho_{pm}$ is the difference between the particle’s and medium’s density and ν the volume), owing to the gravitational force (g), is given by the expression:$$F_{g}=\nu \rho_{pm}g$$

$F_{DEP}$: The DEP is expressed in terms of the difference between the electrode’s potential V (RMS in volts) and $F\left(\vec{r}\right)$, the position in form of a vectorial function [33,47]. The $\varepsilon_{mp}$ is: εmp=εmRe[fCM], where $\varepsilon_{m}\;$ is the permittivity of the medium and Re[f*_CM_*] is the real component of the Clausius-Mossotti’s factor.

$$F_{DEP}=\frac{3}{2}\nu \varepsilon_{mp}V^{2}F\left(\vec{r}\right)$$

$F_{sto}$: The Stokes’ viscous friction opposes the movement; its expression for a spherical cell is:$$F_{sto}=6\pi \eta Rv_{p}$$
where *R* is the cellular radius, *η* is the medium’s dynamic viscosity, and $v_{p}$ represents the particle velocity [48].

Due to its small value, the mass by acceleration term is considered negligible in relation to the other force values:$$F_{sto}-F_{g}-F_{DEP}=0$$

Substituting the $F_{sto}$, $F_{g}$ and $F_{DEP}$ results in the following velocity expression along the *y* axis (Figure 1):(1)$$v_{p}=-1\left(F_{1,EX}V^{2}y^{-3}+v_{s}\right)$$

Thus, Equation (1) shows the theoretical model underpinning this article and is deduced in [33]. 

Where $v_{s}=\frac{2R^{2}\rho_{pm}g}{9\eta }$ is the sedimentation velocity and
(2)$$F_{1,EX}=k\left(\omega \right)F_{1}$$
is the experimental measurement of the dielectrophoretic force, k(ω) is a correction factor introduced to adjust the polarisation of the electrodes [32], $F_{1}$ is a parameter which combines all the dielectric properties of the cell, *V* (RMS in volts) is the difference between the electrodes’ potential, *y* is the position along the Y-axis of the chamber. As in [33] *y* is the axis of symmetry between the two electrodes. Parallel to the gravity force and originated in the imaginary cut-off point of the two V-shaped electrodes (Figure 1).

## 3. Materials and Methods

### 3.1. Electron Microscopy

To visualise the damage caused in the cells by the thermal treatment, a preparation was made for electron microscopy using the cryofracture technique. Scanning electron microscopy techniques (SEM) needs high cell concentrations. With that aim in mind, the samples of viable and non-viable yeast cells were centrifuged at 2100 rpm for 1 min. Another aliquot of both samples was cryofractured to see the difference between both samples. Just before visualisation, a thin layer of gold was deposited on the surface of the cells.

### 3.2. Measurement of Cell Viability

Cell viability was determined through microscope observation employing methylene blue dye according to the Pierce Method [49]. A fraction of cellular suspension was mixed with diluted methylene blue (0.01% methylene blue in 0.3 M of mannitol). The heat-killed yeast cells were blue-dyed and, conversely, the living cells were not blue-dyed [50]. The dead cells presented a blue-dyed cytoplasm. At least 1000 cells were observed [49] whether their cytoplasm was blue-dyed or not. An inverted microscope (Olympus CK40 Tokyo, Japan) was used.

### 3.3. Dielectrophoretic Device

The dielectrophoretic device [32] was made of two gold plated silver electrodes of 5 × 20 × 2 mm^3^ positioned at an angle of 53.13° and with a minimum separation of 90.9 μm between them. This vertical plane configuration allowed the cells to displace without contacting the crystals that confine the solution and avoid being affected by the electric field originated from the electrodes’ edges. Optical fibre was used to light the microscope to prevent heating the sample. A sinusoidal, 30 V peak-to-peak, signal was applied through an AC Tektronix-CFG280 (Beaverton, Oregon, USA). This function generator is capable of generating a signal from 10 kHz to 10 MHz. The signal was monitored using a digital Tektronix TDS 320 (Beaverton, Oregon, USA) oscilloscope (100 MHz, 50 Ms/s).

As described in [33], the cells must follow Equation (1). One regression per cell were made. Therefore, the differences between regressions, depends on:The experiment’s execution: different days, different solutions, etc.All cells in a culture are different (e.g., size, different microenvironments...).Real cells differ from the theoretical model. For example, the number of cells at the electrodes (distorting the field perception), different speed at the time of measurement.

All the three points have been tackled as in [33]. Firstly, the experiment’s execution and differences in the preparation between cells turned out to be not statistically relevant through the ANOVA analysis. The experiment was three-fold repeated. As in the alive ones [33], every time an experiment finishes, the chamber was dismantled and cleaned.

In the same way, the third point was solved as in [31,32,33], only those with a linear regression coefficient higher than 0.98 and a sedimentation velocity located between –3 μm/s and 3 μm/s were used.

In order to compare the values of *F_1_* of this experiment to the ones obtained in [33], the electrodes’ polarisation is to be eliminated. Polarisation of the device electrodes was quantified as described in [31,48]. This is how the k(ω) factor was measured, needed for $F_{1,EX}$ in the Equation (2) calculation and replaced in Equation (1).

### 3.4. Cellular Suspension

A uniform strain (RS-16) of *Saccharomyces cerevisiae* (yeast cells) was used. To obtain cells in a similar growing state, the following method was employed: cells were grown in aqueous medium with 1% of yeast extract, 2% of peptone and 2% of glucose at 28 °C in an agitator incubator at 200 rpm. The cells were collected after 48 h at the end of the exponential-growing phase; the cell population growth was determined by absorbance at 660 nm. The cell suspension was cleaned and resuspended three times in 0.3 M mannitol by 2100 rpm centrifugation for 1 min [33].

A Thoma chamber was used to measure the cellular concentration, which was adjusted to 5000 cells/mm^3^, which is close to the 0.1% imposed by the theoretical model [32]. This concentration is equivalent to a maximum volume percentage value of 0.075%, which is close to 0.1%, the imposed theoretical condition. Moreover, because the cellular velocity is high in the measurement area, the effective cell concentration is lower than the maximum. The suspension was left for an hour to let the cells reach the laboratory temperature, 24 °C.

The heat treatment for non-viable cells was as follows: the aliquot of adjusted cell solution was placed in the test tube in water at 90 °C for 20 min [51]. Afterwards, conductivity of the medium was determined using a Crison CDTM-523 conductimeter at 3.8 kH and adjusted at 4 mS/m. Low conductivity was selected to avoid the Joule effect, that could create convection currents; these currents are able to overshadow the action of DEP [33].

### 3.5. Experimental Procedure

The cellular suspension was loaded into the disconnected chamber of the device after be sealed with vacuum grease. From there on, the chamber was positioned vertically in an optical microscope (Olympus CK40 Tokyo, Japan). Amplitude (15 V) and frequency (0.03, 0.05, 0.075, 0.1, 0.4 and 1 MHz) on the electrodes were both selected previously. It took a few minutes for the cellular lumps to fall, after which the cellular movements were recorded on VHS video. One frequency per experiment and less than five minutes for each experiment.

In line with our previous work [33], all cells came from the same preparation, and each one was measured in the same experiment. Each experiment was repeated thrice. The only measured cells were those which flowed through the *y*-axis of the dielectrophoretic device [33].

Particle motion was recorded using a 1/2” video-camera (SONY) model SSC-C531, with a sensitive area of 6.3 × 4.7 mm, corresponding to a resolution of 500 × 582 pixels [32]. The microscope was focused on the middle Z-axis, middle thickness of the preparation, to eliminate the area near to the glasses that confined the chamber. A computer-generated reticule was superimposed onto the recording. It was marked in 9.09 μm increments with a frame counter at a recording speed of 50 frames per second. The visual area of the microscope was 30 divisions in length along the vertical direction, ranging from the 15th (*y* = 136.35 μm) to the 45th (*y* = 409.05 μm), in relation with the coordinate origin. The recorded experiments were analysed visually, using the frame-advance mode of the VHS magnetoscope. The division number, being crossed by the cell, and the frame count were both annotated.

To obtain $v_{s}$, from taken data, a linear regression from Equation (1) was performed, and at the point of $v_{p}$ by which the electric field is negligible, the velocity of the particle is equal to $v_{s}$. This happens when $V^{2}$ or $y^{-3}$ are equal to 0. There was not a different experiment to measure the sedimentation velocity ($v_{s}$) of Equation (1).

## 4. Results

### 4.1. Electron Microscopy

The heat-killed cells were found to be fractured without any pattern, whereas the control cells were cleanly split through their compartments, as shown in Figure 2.

### 4.2. Cell Viability

Cell viability was determined through microscope observation utilising methylene blue dye according to the Pierce Method [49]. In the alive group, the number of blue-dyed cells was negligible (more than 95% of non-dyed cells) [33]. Nevertheless, in the group of heat-killed cells, more than 95% cells were blue-dyed.

### 4.3. Measurement of the Electrical Properties of the Dielectrophoretic Chamber

According to the theory [32], the electric properties of the chamber were measured to quantify the electrode’s polarisation. The results obtained are shown in Table 1.

Both parameters, parallel resistance and capacitance, depend on the chamber setup. With these values, k(ω) was calculated from Equation (2) as a function of frequency, results are shown in Figure 3.

Thanks to k(ω), we can compare *F_1,EX_* of live cell data [33] with the results of heat-killed cells from this experiment. We can also use the data of conductivities and electrical permittivities of yeast cell compartments found in the literature [46,52].

### 4.4. Cell DEP Measurements

The cells used in the present work, and those analysed in [33], come from the same solution. As the only difference between both groups of cells is the thermal treatment that was applied to the second group, the first group of cells, the cells used in [33], is the alive group to be discussed in this article. The data of the alive group can be seen in [33].

ANOVA showed that there were no statistically significant differences (*p*-values < 0.05) among the three experiments (Figure 4a). Therefore, the three experiments were considered as one (Figure 4b).

From all 279 cells measured, applying the conditions to select cells that follow the theoretical model (linear regression coefficient higher than 0.98 and a sedimentation velocity located between –3 μm/s and 3 μm/s) 88 cells were used, representing 31.5% of the available data. As in the alive ones, there were no statistically relevant differences between the experiments with heat-killed cells. The results for each frequency are shown in Table 2.

In Figure 5, it is shown the experimental data of Table 2 and the theoretical values for heat-killed cells. Theoretical data come from Equation (1) [33], which used the dielectric and geometric parameters of heat-killed yeast cell found in [46,52]. These parameters are shown in Table 3.

## 5. Discussion

To avoid the effect of electrodes polarization between experiments, $F_{1,EX}$ data from heat-killed and live yeast cell populations [33] were compared by using the k(ω) factor (Table 1 and Figure 3). Comparing the results with previous work [33], the $F_{1,EX}$ measured in heat-killed cells was much lower. In all cases of heat-killed cells, the value of $F_{1,EX}$ is below one. Conversely, in alive cells, the value ranges from 3 in the case of 100 kHz and single cells, to 6 for double cells beyond the 1000 kHz. Therefore, the capability of this method to measure changes in alive yeast cells and heat-killed has been validated.

As in [33], three experiments were made (Figure 4a and Table 2). Since there was no statistical difference the three experiments were considered as one (Figure 4b).

In Figure 5, it can be seen that the experimental data obtained in this experiment agrees with cell electrical and geometrical parameters of Table 3. The behaviour exposed in Figure 5 significantly differs from the theoretically expected. It is because there exists distinct, possible, conductivity values, which could make the response vary also. The values taken from bibliography have been acquired with a different strain or cell treatment [46,52].

As shown in the results section, only 31.5% of the available data were used much less than the 78% used in [33]. This might be due to the low velocities which were measured with these cells. As the DEP force ($F_{1,EX}$) decreases with heat treatment, the cell’s experimental speed also decreases. With a lower velocity, it takes more time for the cell to reach the end of the chamber. This has two consequences. First, cells are more susceptible to being influenced by electric or hydraulic effects of the solution’s microenvironment and second, it increases the uncertainty when taking the measure. This is because it becomes difficult to pinpoint when the cell has crossed the line of the grid. This result coincides with the alive cells [33], where the lower the rate, the lower the percentage of cells that met the two requirements to be used as results.

As explained in [46], with the thermal treatment that was performed, the cell wall conductivity is increased due to cytoplasm leaks from the inner parts of the cell. This leakage is due to the fact that the cell membrane breaks. On its way out, cytoplasm is captured by the cell wall mucopolysaccharides, increasing the cell-wall conductivity. On the other hand, the membrane breakage increases the cell-membrane conductivity itself. Nonetheless, the method described in this article is not capable of quantifying such an increase. This is because a huge increment in cell membrane conductivity would be needed, up to 10^−3^ S/m, for this effect to be noticed [33]. This value is much higher than the one proposed by other authors, 10^−5^ S/m [46]. Likewise, other authors [53,54], through electroporation, propose values from 10^−4^ S/m^2^ to 4.3 × 10^−4^ S/m^2^ for the increased membrane conductance originated after the electroporation. These measures are to be considered because despite electroporation is an entirely distinct technique from heat treatment, its outcome over the structure of the cell membrane is almost the same that the one reached through heat treatment. In fact, because they both have the same effect over the cell’s structure, they both explain the increase in cell membrane conductivity through the same principle [41].

In Table 3, data of relative permittivity and geometric features of the cell have been taken from [46,52]. Values of conductivities have been taken from [46]. Therefore, in order to obtain the theoretical model of the heat-killed cells, it was decided to vary from the alive group model [33] the values of cytoplasm conductivity and cell wall conductivity. Concretely, cytoplasm conductivity was changed from 0.6 S/m in alive cells to 0.08 S/m in heat-killed cells. Moreover, cell wall conductivity was modified from 0.0011 S/m in alive cells to 0.011 S/m in heat-killed cells. Such a variation is very similar to the one described by [46]. These authors found by electrorotation a change in cytoplasm conductivity from 0.55 S/m in alive cells to 0.01–0.08 S/m in the heat-killed ones. Additionally, cell wall conductivity varied from 0.01 S/m in alive cells to 0.03 S/m in the heat-killed ones.

Figure 5 has been obtained with the changes mentioned above introduced into the theoretical model. We noticed that the curve of heat-killed yeasts for 30 kHz is almost zero. It makes sense accordingly with the experimental data because, at such frequency, no heat-killed cell could be measured due to its negligible velocity. Nevertheless, the velocity of alive cells was measured.

However, the thesis and values are supported in other publications. [55] offers a value for cell membrane conductivity of 1.6 × 10^−4^ S/m for heat-treated cells and 2.5 × 10^−7^ S/m for viable cells. It means an increase of 1.6 × 10^−4^ S/m approximately, which is closer to the value proposed in this paper than the one shown in [46]. In addition, other authors [56] explain the phenomenon of the decrease in cytoplasm conductivity because of a loss of ions. Such a loss is explained by the cell membrane leakage commented above. This layer behaviour has been studied not only for yeast cells but for Chinese hamster ovary cells too [56] in a temperature spectrum ranged from 7 °C to 50 °C. Surprisingly, the effect is inverted for low temperatures [57] and cell membrane conductivity is decreased.

As shown in Figure 2 in the SEM photographs without cryofracture, the heat-treated cells suffered a loss of form (Figure 2b) versus the alive cells (Figure 2a). In the cells subjected to cryofracture, it can be observed that their rupture is more heterogeneous (Figure 2d,f) in those subjected to thermal treatment than in those of alive cells (Figure 2c,e), which break more homogeneously, revealing their cellular compartments. Moreover, the methylene blue according to the Pierce Method corroborated the hypothesis that the cell membrane is damaged. SEM images do not pretend to bring any conclusion. They are just a tool to demonstrate that the variations upon the cell wall have been made. Therefore, changes in cell properties have been reflected through three different approaches: SEM, Pierce Method and the proposed method in this article.

## 6. Conclusions

This work has been performed making use of two distinct groups of cells, a group of heat-killed cells and another group of alive cells. The exposed method in this article is capable of distinguishing between cells of both groups.

Using Equation (2) is a useful tool to compare experiments with different polarizability electrodes. Measuring this parameter allows us to quantify the real electric field inside the solution.

The experimental values obtained agree with the ones found in [46,52] and they can be explained by a decrease in cytoplasm conductivity and an increase in cell wall conductivity. These variations are due to the formation of pores in the cell wall and membrane, or their breaking. This weakening can be observed using electron microscopy over the cryofractured sample and methylene blue according to the Pierce Method.

Therefore, in the present article it is shown that the used experimental method and theoretical model not only can distinguish cells of different shapes, but also discriminate between cells with different biological features reflected in their electric parameters. In this case, such biological features were the damages caused by a heat shock over the cells. Despite the fact that we used the presented method in our previous publications [31,32,33], all of those works were based on geometric cell properties.

There are other, more developed, alternatives used for cell discrimination, such as electrorotation. Nonetheless, it is the first time in which both Stokes’ Drag Force and DEP are combined to distinguish between biologically differentiated cells. It is the novelty of the studied method, which opens the door for many future works in which improve the efficiency of the presented one, giving rise to a mature methodology. Additionally, the advantage of this method over others is that it can be easily automated and carried out continuously.

## Figures and Tables

**Figure 1 sensors-19-05304-f001:**
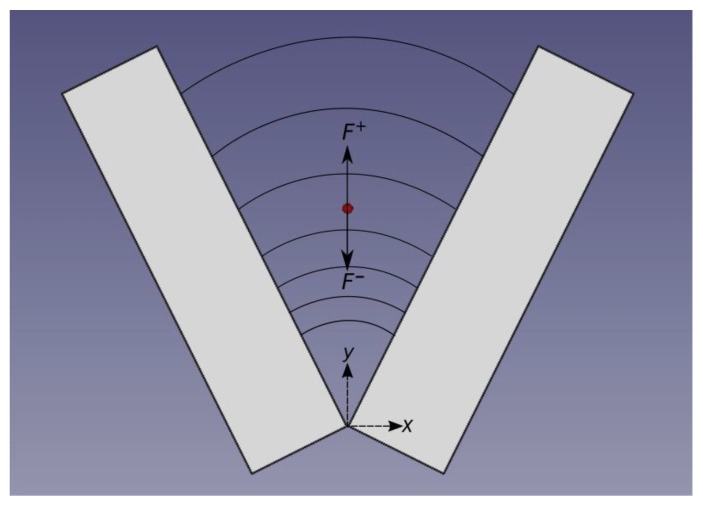
Representation of the experimental setup the red dot symbolizes a yeast cell, displacing along the *y*-axis, the symmetry axis of the chamber. The electric field is represented by lines between electrodes. The electric field intensity is increased along the *y*-axis. *F*^+^ represents the force along the positive direction of *y*-axis that is the Stokes’ Drag Force (Fsto) and *F*^−^ the forces along the negative direction of *y*-axis that are: the force of gravity (F_g_) and the dielectrophoretic force (F_DEP_). Electrode dimensions were 5 × 20 × 2 mm^3^.

**Figure 2 sensors-19-05304-f002:**
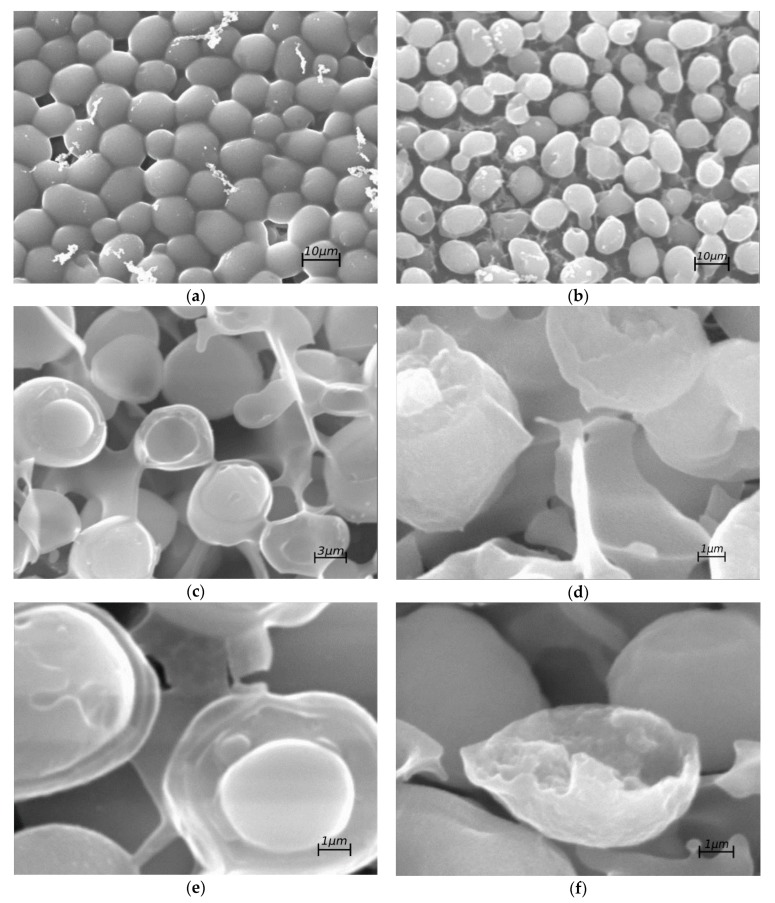
Electron microscopy photographs. Pictures (**a**,**b**) have a magnification of 3500×, (**c**) a magnification of 5000× and (**d**–**f**) of 7000×. Pictures (**a**,**c**,**e**) belong to the alive group and pictures (**b**,**d**,**f**) belong to the group of heat-killed cells.

**Figure 3 sensors-19-05304-f003:**
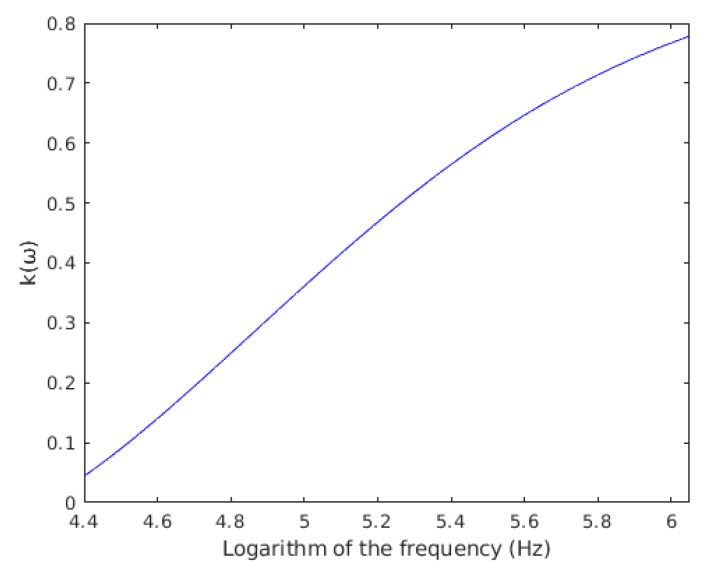
k(ω) as a function of frequency logarithm.

**Figure 4 sensors-19-05304-f004:**
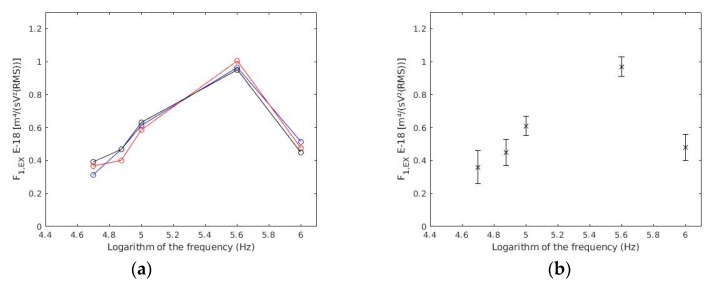
(**a**) *F_1,EX_* as a function of frequency for each experiment. Each colour represents one experiment. (**b**) Results of the three experiences together.

**Figure 5 sensors-19-05304-f005:**
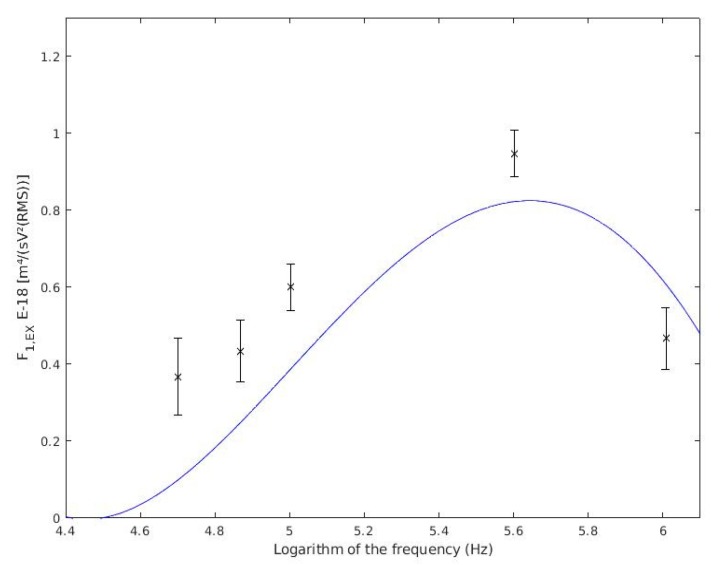
For each experience, the black cross represents the mean of experimental data for $F_{1,EX}$ for heat-killed cells. All the experimental data is within the interval closed by the confidence intervals (two times the standard error). The blue line represents the theoretical values from Equation (1) with electric parameter data of [46].

**Table 1 sensors-19-05304-t001:** Electric properties of the dielectrophoretic chamber.

Parallel resistance (Ω)	3000
Parallel capacitance (F)	$1.2\times 10^{-11}$

**Table 2 sensors-19-05304-t002:** Experimental results of $F_{1,EX}$ for each frequency.

Frequency (log(Hz))	$F_{1,EX}10^{-18}\;$ $\left(\frac{m^{4}}{V^{2}\left(rms\right)S}\right)$	Standard Error $\times 10^{-18}$	Selected Cells	Measured Cells
4.699	0.36	0.05	12	21
4.875	0.45	0.04	16	72
5.000	0.61	0.03	29	69
5.602	0.97	0.03	31	64
6.000	0.48	0.04	14	53

**Table 3 sensors-19-05304-t003:** Data employed to calculate the theoretical values of $F_{1,EX}$ [46,52].

**Conductivities (S/m)**	
Medium	4 × 10^−3^
Cell wall	1.1 × 10^−2^
Cell membrane	0
Cytoplasm	0.008
**Relative permittivity**	
Medium (ε_m_)	77
Cell wall	60
Cell membrane	5.2
Cytoplasm	58
**Geometric parameters (m)**	
Cell radius (R)	3.25 × 10^−6^
Cell wall thickness	0.25 × 10^−6^
Cell membrane thickness	7 × 10^−9^

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
