# Peer review of "Thermal Shock Response of Yeast Cells Characterised by Dielectrophoresis Force Measurement"

_sensors, 2019, doi:10.3390/s19235304_

Round 1
Reviewer 1 Report
The manuscript "Thermal shock response of yeasts cells characterised by dielectrophoresis force measurements" shows the use od DEP to detect damage in yeast cells treated by thermal shock. The document is quite interesting and as I can understand it expands the results from a previous report.
The document needs to be improved.
1) The document is quite short, only 12 pages. The introduction section must to be improved. To me there are a lot of cited reference that are not properly described, i.e. 31 cited references; are mentioned in only two brief paragraphs; however, what are the importance of these references to the document? It is not valid to reference multiple cites [4-12] if there s not really an association with the aim of the manuscript submitted by the authors, and if it is really and association the authors need to give more detail of the cited reference.
2) line 44 typo "d ecades"
3) Line 67, what authors consider a phisiological criteria?
4) Line 129, describe what is uniform strain?
5) Line 163, the symbol of celsius degrees is bizarre, the underline below the circle
6) Figure 1, please homogenize the size of the pictures.
7) Figure 3, Include in the label of this figure the mathematical function describing the theoretical values (include those values), blue line.
8) Line 238, Cell viability was measured through the me.... was already described in the methodology.
Discussion section, the discussion is performed on two cited references! It is necessary to include other kind of reports that considers the physiological state of the microorganisms. Authors must let perfectly clear for the reader what are the physiological states they detected with their study.
Line 276, please replace "In Figure 3, it can be seen that the experimental data obtained in this experiment is the same in shape with those in the literature [51]" to "In Figure 3, it can be seen that the experimental data obtained in this experiment agrees with those reported in [51]"
Line 282 specify what are both types?
The abstract and the conclusion sections, must be improved to reflex the findings of the document
Author Response
REVIEWER 1
The authors thank all the reviewers’ comments. It is sure they will make of our work a better one.
To facilitate the correction, a Word file with tracking changes and a PDF with the final version have been attached to this answer.
The present work is the continuation of another published in this journal:
García-Diego, F.-J.; Rubio-Chavarría, M.; Beltrán, P.; Espinós, F.J. Characterization of Simple and Double Yeast Cells Using Dielectrophoretic Force Measurement. Sensors 2019, 19, 3813.
At the time in which we sent for revision the mentioned work, the reviewers indicated that it would be of great interest if we could perform the same experiment in order to differentiate among cells in different biological states, like living and dead cells.
That experiment was already completed at the moment of our previous work publication. These data were not published just because we thought that the article we were going to publish would be too bigger and difficult to interpret with more content.
After the reviewers exposed their opinion, the authors were encouraged to start a new article within the data they did not published before, the data shown in the present article. That is why the writing of this article is, for us, more difficult than usual.
It must be taken into account that the control population of cells in this article is the group which offered the results of our previous article. Therefore, in the discussion we can only compare our results with the ones already published in the quoted article because there are no more works that implement the method studied in this two articles.
If we want to contrast our results with those of other authors we are to use other methodologies, like electrorotation, more extended to compare in the discussion.
We hope the work to be more relevant and understandable for the readers with the changes we have made.
The manuscript "Thermal shock response of yeasts cells characterised by dielectrophoresis force measurements" shows the use of DEP to detect damage in yeast cells treated by thermal shock. The document is quite interesting and as I can understand it expands the results from a previous report.
Thank you very much for opining that our work is interesting and, indeed, it assumes a methodology to date published in this journal:
García-Diego, F.-J.; Rubio-Chavarría, M.; Beltrán, P.; Espinós, F.J. Characterization of Simple and Double Yeast Cells Using Dielectrophoretic Force Measurement. Sensors 2019, 19, 3813.
As it has been stated previously, and so as to this said by the editor to be understood, the prior work is frequently quoted in the current.
The document needs to be improved.
1) . The document is quite short, only 12 pages. The introduction section must to be improved. To me there are a lot of cited reference that are not properly described, i.e. 31 cited references; are mentioned in only two brief paragraphs; however, what are the importance of these references to the document? It is not valid to reference multiple cites [4-12] if there s not really an association with the aim of the manuscript submitted by the authors, and if it is really and association the authors need to give more detail of the cited reference.
To solve this, the introduction, discussion and conclusion has been completely modified.
One more figure has been added and a theory section has been added.
2) Line 44 typo "d ecades"
Modified
3) Line 67, what authors consider a phisiological criteria?
To avoid talking about cellular physiological criteria that is very ambiguous. The whole paragraph has been rewritten. All “physiological” have been rewriter along the hole text.
“Nonetheless, different cells may present similar shapes but differ in other biological features such as inner cell structure or cell wall composition. Such features are represented in dielectrophoretic cell properties and, as a consequence, are reflected in the cell’s DEP response. In this work, thermally treated (heat shock) of yeast cells are studied. This heat shock supposes changes in electric parameters in cell wall conductivity, cell membrane conductivity, cell membrane permittivity with respect the cells of the non-treated group [33]. These changes provoke a distinct reaction to DEP force, conferring different measured velocities in cells.
Therefore, these features constitute potential DEP-based criteria for cell diferentiation. According to it, among the biologically-oriented cell differentiation methods there is none based on DEP measurement through direct application of Newton’s Second Law and Stokes drag force. This article aims to fill this gap. We demonstrate that, despite being based on geometric cell characteristics [31-33], this method is valid for the differentiation of different cell biological states.”
4) Line 129, describe what is uniform strain?
To avoid this mistake, the sentence:
“Saccharomyces cerevisiae (yeast cells) from a uniform strain were used.”
Was changed to
“A uniform strain of Saccharomyces cerevisiae (yeast cells) from were used.”
5) Line 163, the symbol of celsius degrees is bizarre, the underline below the circle
They have been modified all places in which these errors appeared.
6) Figure 1, please homogenize the size of the pictures.
Modified
7) Figure 3, Include in the label of this figure the mathematical function describing the theoretical values (include those values), blue line.
To avoid this mistake, the sentence:
“The blue line represents the theoretical values from bibliography”
Was changed to:
“The blue line represents the theoretical values from Equation (1) with electric parameter data of [45]”
8) Line 238, Cell viability was measured through the me.... was already described in the methodology.
To avoid this mistake, the sentence:
“Cell viability was measured through the methylene blue method proposed by [55].”
Was changed to:
“Cell viability was determined through microscope observation utilising methylene blue dye according to the Pierce Method [49] as is stated in the Materials and Methods section.”
Discussion section, the discussion is performed on two cited references! It is necessary to include other kind of reports that considers the physiological state of the microorganisms. Authors must let perfectly clear for the reader what are the physiological states they detected with their study.
The discussion section has been modified and is attached at the end of this answer.
Line 276, please replace "In Figure 3, it can be seen that the experimental data obtained in this experiment is the same in shape with those in the literature [51]" to "In Figure 3, it can be seen that the experimental data obtained in this experiment agrees with those reported in [51]"
Modified
Line 282 specify what are both types?
These both changes have been made.
The abstract and the conclusion sections, must be improved to reflex the findings of the document.
The abstract has been changed to:
Abstract: Dielectrophoresis is an electric force experienced by particles subjected to non-uniform electric fields. Recently, several technologies have been developed focused on the use of dielectrophoretic force (DEP) to manipulate and detect cells. On the other hand, there is no such great development in the field of DEP-based cell differentiation methods. Despite the demand for methods to differenciate biological cell states, most DEP developed methods have been focused on differentiation through geometric parameters. The novelty of the present work relies upon the point that DEP force cell measurement is used as a differentiation method, capable of detecting thermally treated dead yeast cells from non-treated living ones. This is used as an example of different biological state of cells. It comes from the fact that biological properties have their reflection in the electric properties of the particle, in this case a yeast cell. To demonstrate such capability of the method, 279 thermally treated cells were measured and compared with untreated cell data from the literature. For each cell, six speeds were taken at different points in its trajectory. The electric parameters in cell wall conductivity, cell membrane conductivity, cell membrane permittivity of the yeast cell were successfully compared with the bibliography. Finally, non-treated and treated cells were distinguished based on that measures. Our results can be explained through the well known damage of cell structures characteristic of thermally treated cells.
The conclusions section has been changed to:
6. Conclusions
It was started from two different groups of cells. One, a group of thermally treated cells. And second, the control group of non-treated cells. The exposed method is capable of distinguish between cells of both groups. The experimental values obtained agree with the ones found in the bibliography [45] and, moreover, they can be explained by a decrease in cytoplasm conductivity and an increase in cell wall conductivity. These variations are due to the formation of pores in the cell wall and membrane, or their breaking. This weakening can be observed using electron microscopy over the cryofractured sample and methylene blue according to the Pierce Method.
Therefore, in the present article is shown that the used experimental method and theoretical model not only can distinguish cells of different shapes but differentiate between cells with different biological features reflected in their electric parameters. In this case, such biological features were the damages caused by a heat shock over the cells. Despite we used the present method in our previous works, they all were based on geometric cell properties.
There are other, more developed, alternatives used for cell differentiation, such as electrorotation. Nonetheless, it is the first time in which both Stokes’ Drag Force and DEP are combined so as to distinguish between biologically differentiated cells. It is the novelty of the studied method which opens the door for many future works in which improve the efficiency of the presented one, giving rise to a mature methodology. Besides, the advantage of this method over others is that it can be easily automated and carried out continuously.
Reviewer 2 Report
The authors present in this paper a DEP based characterisation methods for yeast study. The measurement of the movement of yeast in an electric field was used to determine their specific properties and compared to SEM observation and optical characterisation.
This article seems to only offer a small addition to previously published work. The authors use the same dataset as control but do not present the control results in this paper. This led to an article which is not self-supported and do not offer enough interest for publications. Moreover, a large number of the claim of this paper in term of novelty, characterisation and conclusion are not supported enough by the experiments.
Before resubmission, the following point should be addressed.
The authors need to provide in this paper all the comparison with previously published data set to offer the reader the possibility to properly judge this work independently. Figure explaining the geometry of the experiment should be added. The author claim to present a new method for yeast live/death assess. But a large number of previously publish study have been using DEP for this specific purpose in fixed and continuous flow system. The authors need to clarify this point. The SEM presented in figure 1 only offer anecdotal evidence of the authors claim. Only one specific cell seems to be selected as a demonstration of the damage on the cells. Statistical analyses of the SEM picture are required to make any conclusion. A full rewrite of the introduction is needed to give more insight on the cited paper. This paper need to be understandable without going through all the references. Additionally, please define the concept that you are presenting. As an example, CTC detection is a type of characterisation. What do you really mean by characterisation in L45? The structure of the paper needs revision to properly separate the theoretical model, the experiment plan, the comparison of the experiments results with the various control and discussion.
Other smaller Comment:
In the abstract, “The results were successfully compared with the bibliography”, please clearly state what metrics you compare and what the indicator of “success” is.
The authors do not explain how the sedimentation velocity of the cells was measured or obtained during the experiments.
L 33-34 “They have great potential and many practical applications”. Please be specific
L36-38 The author should be more selective in their references selections. Citing a relevant review on the topic of DEP should be consider.
L44 “d ecades“, extra space
Author Response
REVIEWER 2
The authors thank all the reviewers’ comments. It is sure they will make of our work a better one.
To facilitate the correction, a Word file with tracking changes and a PDF with the final version have been attached to this answer.
The present work is the continuation of another published in this journal:
García-Diego, F.-J.; Rubio-Chavarría, M.; Beltrán, P.; Espinós, F.J. Characterization of Simple and Double Yeast Cells Using Dielectrophoretic Force Measurement. Sensors 2019, 19, 3813.
At the time in which we sent for revision the mentioned work, the reviewers indicated that it would be of great interest if we could perform the same experiment in order to differentiate among cells in different physiological states, like living and dead cells.
That experiment was already completed at the moment of our previous work publication. These data were not published just because we thought that the article we were going to publish would be too bigger and difficult to interpret with more content.
After the reviewers exposed their opinion, the authors were encouraged to start a new article within the data they did not published before, the data shown in the present article. That is why the writing of this article is, for us, more difficult than usual.
It must be taken into account that the control population of cells in this article is the group which offered the results of our previous article. Therefore, in the discussion we can only compare our results with the ones already published in the quoted article because there are no more works that implement the method studied in this two articles.
If we want to contrast our results with those of other authors we are to use other methodologies, like electrorotation, more extended to compare in the discussion.
We hope the work to be more relevant and understandable for the readers with the changes we have made.
The authors present in this paper a DEP based characterisation methods for yeast study. The measurement of the movement of yeast in an electric field was used to determine their specific properties and compared to SEM observation and optical characterisation.
In effect, in the present paper is studied a DEP based characterisation method. Concretely, it is measured the velocity of the cell within a variable electric field. The exposed method, through the measured velocity, brings us the electric properties of each cell, which in this case we related to biological features of the cell rather than geometric characteristics, as done in previous works with the same metodology. Once the treated and non-treated groups of cells were correctly differentiated, it was used the SEM to observe the poblation. As it was stated, the cells of the treated group not only presented different electric features but also they appeared completely fragmented in SEM images. SEM resulted in an indispensable tool to ensure the effectiveness of the thermal treatment.
Such a demonstration could have been achieved through by the methylene blue but the authors wanted the detail this through SEM. Otherwise, it would have been impossible to visualize the damages upon the cell wall.
To shed light, the following paragraph has been modified:
“As shown in the electron microscopy photographs without cryofracture, the heat treated cells suffered a loss of form (b) versus the controls (a). In the cells subjected to cryofracture, it can be observed that their rupture is more heterogeneous (d, f) in those subjected to thermal treatment than in those of control (c, e), which break more homogeneously, revealing their cellular compartments. Moreover, the methylene blue according to the Pierce Method corroborated the hypothesis that the cell membrane is damaged. Therefore, changes in cell properties have been reflected through three different approaches: SEM, Pierce Method and the proposed in this article.”
This article seems to only offer a small addition to previously published work.
When our the work referenced but the reviewer was sent:
García-Diego, F.-J.; Rubio-Chavarría, M.; Beltrán, P.; Espinós, F.J. Characterization of Simple and Double Yeast Cells Using Dielectrophoretic Force Measurement. Sensors 2019, 19, 3813.
As told before, the reviewers congratulated us because of the novelty and usefulness of the proposed technique to measure the DEP. Until that moment there was employed only another technique called electrorotation. They too encouraged us to probe the validity of the proposed method to measure physiological changes in the cell.
That is the reason why we started this article employing the same technique upon two geometrically-identical cell populations. These two populations presented at the same time different biological features. In order to clarify this point, the following paragraph has been added:
Therefore, in the present article is shown that the used experimental method and theoretical model not only can distinguish cells of different shapes but differentiate between cells with different biological features reflected in their electric parameters. In this case, such biological features were the damages caused by a heat shock over the cells. Despite we used the present method in our previous works, they all were based on geometric cell properties.
The authors use the same dataset as control but do not present the control results in this paper.
Indeed, the control results of this work can be read in our prior article. So as to shed light upon this point, it has been modified the following paragraph in the results section:
“The cells used in the present work and those analysed in [33] come from the same solution. As the only difference between both groups of cells is the thermal treatment that was applied to the second group, the first group of cells, the cells used in [33], is the control group to be discussed in this article. The data of the control group can be seen in [33].”
To avoid a misunderstanding about it, we have modified the introduction in its entirety
This led to an article which is not self-supported and do not offer enough interest for publications.
We believe that with the former paragraph written amd introduction changes should be enough to remove any doubt.
Moreover, a large number of the claim of this paper in term of novelty, characterisation and conclusion are not supported enough by the experiments.
Before resubmission, the following point should be addressed.
The authors need to provide in this paper all the comparison with previously published data set to offer the reader the possibility to properly judge this work independently.
We understand that the reviewer wants us to compare this work with our previous one. This has been made in the discussion section:
“Comparing the results with previous work [33], the F1 measured in thermally treated cells was much lower. In all cases of treated cells, the value of F1 is beneath 1. Conversely, in non-treated cells, the value ranges from 3 in the case of 100 kHz and single cells, to 6 for double cells beyond the 1000 kHz. Therefore, the capability of this method to measure changes in biological cell states has been validated.”
For a better understanding of the work, we have changed the verb "characterize" to "differentiate" since in our work we have not given values of any cellular electrical parameter.
Figure explaining the geometry of the experiment should be added.
This figure has been added. It is the current figure 1.
The author claim to present a new method for yeast live/death assess. But a large number of previously publish study have been using DEP for this specific purpose in fixed and continuous flow system. The authors need to clarify this point.
The introduction and conclusions sections have been modified to make this point clear. Besides, it has been added the sentence:
“In the previous experiment [33], it was verified that this method was sensitive in detecting variations in cell wall conductivity, cell membrane conductivity, cell membrane permittivity and cell shape. It was achieved for both single and double cells. These changes in conductivity and permittivity may be caused by thermal treatments, ultrasounds, or the effect of detergents [41–44].
The aim of this work is to demonstrate that these changes, well documented in bibliography [42, 45], could be measured through the same technique described in [33]. The ease of performing the treatment and the abundance of bibliography [42, 45] were the two main reasons we decided on a thermal treatment among the three possible alterations mentioned above. In this experiment, the same cell preparation and the same chamber were used as in the previous assay [33].”
The SEM presented in figure 1 only offer anecdotal evidence of the authors claim. Only one specific cell seems to be selected as a demonstration of the damage on the cells.
Statistical analyses of the SEM picture are required to make any conclusion.
This point has already been explained. We have not pretended to make such a conclusion. To clarify this point it has been added in the discussion the following sentence:
“SEM images do not pretend to bring any conclusion. They are just a tool to demonstrate that the variations upon the cell wall have been made.”
A full rewrite of the introduction is needed to give more insight on the cited paper. This paper need to be understandable without going through all the references.
To solve this issue we have modified the introduction.
Additionally, please define the concept that you are presenting. As an example, CTC detection is a type of characterisation.
To fulfil this lack of information the following paragraph has been added in the introduction section, and a very similar one in the conclusions.
“The aim of this work is to demonstrate that these changes, well documented in bibliography [42, 45], could be measured through the same technique described in [33]. The ease of performing the treatment and the abundance of bibliography [42, 45] were the two main reasons we decided on a thermal treatment among the three possible alterations mentioned above. In this experiment, the same cell preparation and the same chamber were used as in the previous assay [33].”
What do you really mean by characterisation in L45?
For a better understanding of the work, we have changed the verb "characterize" to "differentiate" since in our work we have not given values of any cellular electrical parameter.
The structure of the paper needs revision to properly separate the theoretical model, the experiment plan, the comparison of the experiments results with the various control and discussion.
We have faced this issue extending the theoretical model section and through many improvements in introduction, discussions, and conclusions sections, many of them have been mentioned before.
Other smaller Comment:
In the abstract, “The results were successfully compared with the bibliography”, please clearly state what metrics you compare and what the indicator of “success” is.
To clarify this point, the exposed sentence has been substituted by:
“The electric parameters in cell wall conductivity, cell membrane conductivity, cell membrane permittivity of the yeast cell were successfully compared with the bibliography.”
The authors do not explain how the sedimentation velocity of the cells was measured or obtained during the experiments.
To make this point clear we have added the following paragraph:
“To obtain , from taken data, a linear regression from Equation (1) was performed, and the point of by which electric field is negligible the velocity of the particle is equal to the . This happens when or are equal 0. There was no a different experiment to measure the sedimentation velocity () of Equation (1).”
L 33-34 “They have great potential and many practical applications”. Please be specific.
To solve this issue we have slightly reduced the amount of quotes and, on the other hand, developed a little all the topics considered. All in the introduction section.
L36-38 The author should be more selective in their references selections. Citing a relevant review on the topic of DEP should be consider.
We have included in bibliography important reviews. Especially:
Pethig, R. Review—Where Is Dielectrophoresis (DEP) Going? J. Electrochem. Soc. 2017, 164, pp. 3049–3055.
But also:
Abd Rahman, N.; Ibrahim, F. and Yafouz, B. Dielectrophoresis for Biomedical Sciences Applications: A Review. Sensors 2017, 17, p. 449.
L44 “d ecades“, extra space.
It has been solved. We strongly regret this kind of mistakes.
Reviewer 3 Report
The findings in this paper are no unique to the chosen DEP monitoring technique. Electrorotation (referenced), height (U. Manitoba) or trapping efficiency-based (Lehigh U.) techniques achieve similar or superior insight for live thermally stressed mammalian cells. In some cases, where DEP is combined with dielectric spectroscopy, internal properties can be determined which are sensitive to conductivity changes following membrane integrity disruption...
[39] is too heavily relied on. Is this an independent work? It is fine to draw some similarities and expand on previous work, but the reader should be put in conditions to appreciate the current work without going through the previous one. If some methods/findings in [39] are so critical to the present material, then they should be briefly summarized.
There is very little information about the thermal treatment.
How was the medium conductivity chosen? Why so low? For how long 95% cell viability is maintained in such a medium?
Fig. 1 should include dimension bars. If the bar for Fig. 1 (a) and (b) is the same one could argue that treated cells appear slightly smaller and maybe the observed DEP response does depend on geometrical alterations.
Language needs revision. Just two issues: line 207 "to diminish the third point"? Line 216: Remove "in".
Author Response
REVIEWER 3
The authors thank all the reviewers’ comments. It is sure they will make of our work a better one.
To facilitate the correction, a Word file with tracking changes and a PDF with the final version have been attached to this answer.
The present work is the continuation of another published in this journal:
García-Diego, F.-J.; Rubio-Chavarría, M.; Beltrán, P.; Espinós, F.J. Characterization of Simple and Double Yeast Cells Using Dielectrophoretic Force Measurement. Sensors 2019, 19, 3813.
At the time in which we sent for revision the mentioned work, the reviewers indicated that it would be of great interest if we could perform the same experiment in order to differentiate among cells in different physiological states, like living and dead cells.
That experiment was already completed at the moment of our previous work publication. These data were not published just because we thought that the article we were going to publish would be too bigger and difficult to interpret with more content.
After the reviewers exposed their opinion, the authors were encouraged to start a new article within the data they did not published before, the data shown in the present article. That is why the writing of this article is, for us, more difficult than usual.
It must be taken into account that the control population of cells in this article is the group which offered the results of our previous article. Therefore, in the discussion we can only compare our results with the ones already published in the quoted article because there are no more works that implement the method studied in this two articles.
If we want to contrast our results with those of other authors we are to use other methodologies, like electrorotation, more extended to compare in the discussion.
We hope the work to be more relevant and understandable for the readers with the changes we have made.
The findings in this paper are no unique to the chosen DEP monitoring technique. Electrorotation (referenced), height (U. Manitoba) or trapping efficiency-based (Lehigh U.) techniques achieve similar or superior insight for live thermally stressed mammalian cells.
The newness of the proposed method lies in the use of the Stokes’ Drag Force. That techniques do not use principle and that is precisely the reason which explains its lack of efficiency compared with other techniques. There is plenty of work to do on this approach.
To explain such issue, the conclusions section has been modified::
“There are other, more developed, alternatives used for cell differentiation, such as electrorotation. Nonetheless, it is the first time in which both Stokes’ Drag Force and DEP are combined so as to distinguish between biologically differentiated cells. It is the novelty of the studied method which opens the door for many future works in which improve the efficiency of the presented one, giving rise to a mature methodology. Besides, the advantage of this method over others is that it can be easily automated and carried out continuously.”
Also, introduction, Conclusions, discussion and the hole MS has been changed to explain this.
In some cases, where DEP is combined with dielectric spectroscopy, internal properties can be determined which are sensitive to conductivity changes following membrane integrity disruption.
Dielectric spectroscopy study cell at different all frequencies but they cannot be reached such measures in a continuous manner. On the contrary, it is true that the mentioned changes can be detected with electrorotation and, consequently, the measures can be obtained. Our technique just follows the same approach, but utilizing other method.
To clarify this point, Introduction, Discussion and conclusions have been written again taking into account this.
As explained before, the conclusions section has been modified:
“There are other, more developed, alternatives used for cell differentiation, such as electrorotation. Nonetheless, it is the first time in which both Stokes’ Drag Force and DEP are combined so as to distinguish between biologically differentiated cells. It is the novelty of the studied method which opens the door for many future works in which improve the efficiency of the presented one, giving rise to a mature methodology. Besides, the advantage of this method over others is that it can be easily automated and carried out continuously.”
[39] is too heavily relied on. Is this an independent work? It is fine to draw some similarities and expand on previous work, but the reader should be put in conditions to appreciate the current work without going through the previous one. If some methods/findings in [39] are so critical to the present material, then they should be briefly summarized.
This point has been answered in the first response to reviewer 3 a few paragraphs above and during the whole letter in many replies.
There is very little information about the thermal treatment.
This has been solved in discussion section with the following paragraph:
“However, the thesis and values are supported in other publications. [51] offers a value for cell membrane conductivity of S/m for heat-treated cells and S/m for viable cells. It means an increase of S/m approximately, which is far more close to the value proposed in this paper than the one offered in [45]. In addition, other authors [52] explain the phenomenon of the decrease in cytoplasm conductivity because of a loss of ions. Such a loss is explained by the cell membrane leakage commented above. This layer behaviour has been studied not only for yeast cells but for chinese hamster ovary cells too [52] in a temperature spectrum ranged from 7 °C to 50 °C. Surprisingly, the effect is inverted for low temperatures [53] and cell membrane conductivity is decreased.”
How was the medium conductivity chosen? Why so low?
To explain this, the following paragraph has been added:
“Low conductivity was achieved to avoid the Joule effect. The Joule effect could create of convection currents; these currents are able to overshadow the action of DEP [33].”
Fig. 1 should include dimension bars.
They have been included
If the bar for Fig. 1 (a) and (b) is the same one could argue that treated cells appear slightly smaller and maybe the observed DEP response does depend on geometrical alterations.
We hope that the newly added bars solve this problem.
Language needs revision. Just two issues: line 207 "to diminish the third point"?
It has been corrected.
Line 216: Remove "in".
It has been done.
Round 2
Reviewer 1 Report
Spscific comments to the manuscript: Thermal shock response of yeast cells characterised by dielectrophoresis force measurement, sensors-620286.
I am glad the authors have taken into account the previous suggestions; however, there are additional queries that authors must fulfill.
Abstract:
Line 26: Authors state that cell wall conductivity, cell membrane conductivity, cell membrane permittivity of the yeast cell were successfully compared with the bibliography; however, in this manuscript there are not associated results to these variables. Please, explain.
I consider there is an overuse of the word differentiation/different in the manuscript, i.e. Line 97, differentiation of different....
Line 128, Revise the bizarre symbol @mp,
Line 185, Once again, what is a uniform yeast strain, it is better to denote the type of strain, i.e., ATCC.
Line 192, how do the authors guarantee that enzymes were kept constant during cultivations? What type of enzymes were measured and what techniques were used? Is this relevant for the experiments?
Results section: In my opinion there is not enough information to support the results of the experiment. This section is quite short. For instance, the half of the electron microscopy subsection must be located in the methodology section, "cryofracture technique".
There is not a proper discussion of the results of the manuscript. What are the relevance of Table 1, 2 and Figure 3?
Lines 264-270 seems to me discussion, more than results.
Table 4, include the values of cell wall conductivity, cell membrane conductivity, cell membrane permittivity and correlated them to thermal treatment.
Line 276: Authors state: There were no statistically relevant differences between the experiments with thermally treated cells, please show those statistical results. These results apply to Figure 4? it seems that in this case there are significant differences between the experimental and theoretical values?
Line 310. Authors state that with a lower velocity, it takes more time for the cell to reach the end of the chamber. What are the factors that affect the velocity of the cell in the chamber?
Line 355, please specify what biological changes were validated?
Line 394, E. coli, must be written in italics.
Only 5 references are used in the discussion section, the same used in the previous paper reported in [33].
Author Response
The authors appreciate the excellent review by the referees. We believe that the work has improved greatly since its first review.
The revised work is sent with change control in “*.dox” file and another without change control in “*.pdf” for better reading.
The most noticeable changes is that the order of the experiments of materials and methods has been modified. The word "differentiation" has been changed to "discrimination" as advised by a referee. Cells not treated thermally by "alive cells" and those thermally treated by "heat-killed" cells.
The discussion and conclusions have been rewritten.
We believe that all these changes have made our MS much more readable.
regards
Line 26: Authors state that cell wall conductivity, cell membrane conductivity, cell membrane permittivity of the yeast cell were successfully compared with the bibliography; however, in this manuscript there are not associated results to these variables. Please, explain.
The sentence in Line 26 has been changed:
“The electric parameters in cell wall conductivity, cell membrane conductivity, cell membrane permittivity of the yeast cell from bibliography explains the DEP experimental force measured.”
I consider there is an overuse of the word differentiation/different in the manuscript, i.e. Line 97, differentiation of different....
Yes, it's correct. Also "Cell differentiation" is confusing since it could be interpreted as the process where a cell changes from one type to another. We have used "cell discrimination" instead. And change “differentiation” for “discrimination” in all sentences.
Line 128, Revise the bizarre symbol @mp,
It has been fixed.
Line 185, Once again, what is a uniform yeast strain, it is better to denote the type of strain, i.e., ATCC.
The sentence in Line 185 has been completed:
"A uniform strain (RS-16) of Saccharomyces cerevisiae (yeast cells) were used"
Line 192, how do the authors guarantee that enzymes were kept constant during cultivations? What type of enzymes were measured and what techniques were used? Is this relevant for the experiments?
In this paragraph we want to point out that the cultivation of yeasts was equal to that of our previous work [33]. As the referee has supposed, in our experiment the enzymes really are not important since it is not the intention of the authors to differentiate yeasts with different forms of growth. That is why we have eliminated that sentence since it did not contribute anything.
In my opinion there is not enough information to support the results of the experiment. This section is quite short.
We agree that the results section is short for what we are used to be published today. But we believe results are relevant since measurements with this technique of these cellular parameters have never been published. Therefore we believe that the results are a novelty.
To expand this section, the results of the three experiments have been added, also Figure 4 with this results.
For instance, the half of the electron microscopy subsection must be located in the methodology section, "cryofracture technique."
The sentence:
“To visualise the damage caused in the cells by the thermal treatment, a preparation was made for electron microscopy using the cryofracture technique”.
has been moved to the methodology section and that section has been reorganized
There is not a proper discussion of the results of the manuscript. What are the relevance of Table 1, 2 and Figure 3?
To solve this issue it has been added the following paragraph in discussion:
"To avoid the effect of electrodes polarization between experiments, data from heat-killed and live yeast cell populations [33] were compared by using the factor (Table 1 and Figure 3). Comparing the results with previous work [33], the measured in heat-killed cells was much lower. In all cases of heat-killed cells, the value of is below one"
And, the following paragraph
"As in [33] three experiments were made (Figure 4a. and Table 2.). Since there was no statistical difference the three experiments were considered as one (Figure 4b)."
Lines 264-270 seems to me discussion, more than results.
We agree that they are not results. We believe they are “methods” because there is no discussion about this method. Only a paper is cited because we use the same method.
This paragraph has been moved to methods
Line 276: Authors state: There were no statistically relevant differences between the experiments with thermally treated cells, please show those statistical results.
Now we sent with the work the statistical results. It has been added the Figure 4 in which such results are shown.
These results apply to Figure 4? it seems that in this case there are significant differences between the experimental and theoretical values?
In the new text Figure 4 is Figure 5
In effect, so as to correct make it clear we are added the following paragraph in the discussion section:
"In Figure 5, it can be seen that the experimental data obtained in this experiment agrees with cell electrical and geometrical parameters of Table 3. The behaviour exposed in Figure 5 significantly differs from the theoretically expected. It is because they exist distinct, possible, conductivity values, which could make the response to vary also, the values taken from bibliography have been acquired with different strain or cell treatment."
Line 310. Authors state that with a lower velocity, it takes more time for the cell to reach the end of the chamber. What are the factors that affect the velocity of the cell in the chamber?
The discussion has been ordered.
To explain the decrease in speed the following sentence has been added:
"As shown in the results section, only 31.5 % of the available data were used much less than the 78% used in [33]. This might be due to the low velocities which were measured with these cells. As the DEP force () decreases with heat treatment, the cell's experimental speed also decreases."
Line 355, please specify what biological changes were validated?
The sentence in Line 355 has been changed:
"Therefore, the capability of this method to measure changes in alive yeast cells and heat-killed has been validated."
Line 394, E. coli, must be written in italics.
Solved
Only 5 references are used in the discussion section, the same used in the previous paper reported in [33].
We have fixed this issue through the quotation of 4 authors more. Not only their opinions but we have compared their values to the ones offered in this article.
[41] Li, H.; Multari, C.; Palego, C.; Ma, X.; Du, X.; Ning, Y.; Buceta, J.; Hwang, J. C. M. and Cheng, X. Differentiation of live and heat-killed E. coli by microwave impedance spectroscopy. B: Chemical. 2018, 255, pp. 1614–1622.
[52] Asami, K. and Yonezawa, T. Dielectric Behavior of Wild-Type Yeast and Vacuole-Deficient Mutant Over a Frequency Range of 10 kHz to 10 GHz, Biophysical Journal, 1996, 71, 2192-2200.
[53] Pavlin, M. and Miklavčič, D. Effective Conductivity of a Suspension of Permeabilized Cells: A Theoretical Analysis. Biophysical Journal, 2003, 85, pp. 719–729.
[54] Hibino, M.; Shigemori, M.; Itoh, H.; Nagayama, K. and Kinosita K. Jr. Membrane conductance of an electroporated cell analyzed by submicrosecond imaging of transmembrane potential. Biophysical Journal. 1991, 59, pp. 209–220.
Reviewer 2 Report
The authors have done an extensive work to alleviate the concern of the first round of review, the link to their previous work has been addressed and a large number of information has been added.
I advices those minor correction and addition as well as a proper grammar checking of the whole manuscript before publication.
Figure 1 : Add scale bar to the schematic (or at least electrodes dimension in the figure caption) L185 : Missing word or reference “…Saccharomyces cerevisiae (yeast cells) from ??? were used.” L204 “Low conductivity was achieved to avoid the Joule effect” Replace “Achieved” by “selected” or another synonym L361 “It was started from two different groups of cells. One, a group of thermally treated cells. And second, the control group of non‐treated cells. The exposed method is capable of distinguish between cells of both groups.” This need to be rephrased in a grammatically structured way (all sentences need verb) L374 : “Despite we used the present method in our previous works, they all were based on geometric cell properties.” Review the grammar
Author Response
The authors appreciate the excellent review by the referees. We believe that the work has improved greatly since its first review.
The revised work is sent with change control in “*.dox” file and another without change control in “*.pdf” for better reading.
The most noticeable changes is that the order of the experiments of materials and methods has been modified. The word "differentiation" has been changed to "discrimination" as advised by a referee. Cells not treated thermally by "alive cells" and those thermally treated by "heat-killed" cells.
The discussion and conclusions have been rewritten.
We believe that all these changes have made our MS much more readable.
regards
Figure 1: Add scale bar to the schematic (or at least electrodes dimension in the figure caption).
Electrode dimensions have been added in figure caption:
"Figure 1. Representation ......................................................................... ................Electrode dimensions were 5x20x2 mm."
Line 185: Missing word or reference “…Saccharomyces cerevisiae (yeast cells) from ??? were used.”
The sentence has been completed, sorry for the confusion:
“A uniform strain (RS-16) of Saccharomyces cerevisiae (yeast cells) were used.”
Line 204: “Low conductivity was achieved to avoid the Joule effect” Replace “Achieved” by “selected” or another synonym.
The sentence has been changed
"Low conductivity was selected to avoid the Joule effect, that could create convection currents; these currents are able to overshadow the action of DEP [33]."
Line 361: “It was started from two different groups of cells. One, a group of thermally treated cells. And second, the control group of non‐treated cells. The exposed method is capable of distinguish between cells of both groups.” This need to be rephrased in a grammatically structured way (all sentences need verb) .
This paragraph has been completely restructured and reallocated.
Also, to avoid this, cells not treated thermally was changed to by "alive cells" and those thermally treated by "heat-killed" cells.
Line 374: “Despite we used the present method in our previous works, they all were based on geometric cell properties.”
That sentence has been replaced by the following:
"Despite we used the presented method in our previous publications [31-33], those all works were based on geometric cell properties."
Review the grammar
Certainly, we are non-native English speakers and have problems when writing. Therefore, a native who has always met our expectations has reviewed the grammar in the first version.
A native did not review the changes made in this revised version. We have done all the best we can.
The magazine "Sensors" makes a grammar review before publishing.
I think this will be enough, but if the editor considers it, we will take back the MS to the grammar reviewer.
Reviewer 3 Report
"Cell differentiation" is confusing since it could be interpreted as the process where a cell changes from one type to another. Please use "cell discrimination" instead.
Please reference Lehigh U's work on dielectric spectroscopy based discrimination of heat-killed E. coli also using DEP trapping: https://doi.org/10.1016/j.snb.2017.08.179
Author Response
The authors appreciate the excellent review by the referees. We believe that the work has improved greatly since its first review.
The revised work is sent with change control in “*.dox” file and another without change control in “*.pdf” for better reading.
The most noticeable changes is that the order of the experiments of materials and methods has been modified. The word "differentiation" has been changed to "discrimination" as advised by a referee. Cells not treated thermally by "alive cells" and those thermally treated by "heat-killed" cells.
The discussion and conclusions have been rewritten.
We believe that all these changes have made our MS much more readable.
regards
"Cell differentiation" is confusing since it could be interpreted as the process where a cell changes from one type to another. Please use "cell discrimination" instead.
Indeed, you are absolutely right, we have changed this word throughout the text and we believe that it has greatly improved the understanding of all the work.
However, there are works that do speak of “differentiation”. But as you told it's confusing (Differentiation of live and heat-killed E. coli by microwave impedance spectroscopy)
Please reference Lehigh U's work on dielectric spectroscopy based discrimination of heat-killed E. coli also using DEP trapping: https://doi.org/10.1016/j.snb.2017.08.179
Great reference, it has been referenced in both the introduction and the discussion sections.
Round 3
Reviewer 1 Report
All my queries have been fulfilled.